# Photonic Integrated Interrogator for Monitoring the Patient Condition during MRI Diagnosis

**DOI:** 10.3390/s21124238

**Published:** 2021-06-21

**Authors:** Mateusz Słowikowski, Andrzej Kaźmierczak, Stanisław Stopiński, Mateusz Bieniek, Sławomir Szostak, Krzysztof Matuk, Luc Augustin, Ryszard Piramidowicz

**Affiliations:** 1Institute of Microelectronics and Optoelectronics, Warsaw University of Technology, 00-662 Warsaw, Poland; andrzej.kazmierczak@pw.edu.pl (A.K.); stanislaw.stopinski@pw.edu.pl (S.S.); mateusz.bieniek3.stud@pw.edu.pl (M.B.); s.szostak@elka.pw.edu.pl (S.S.); ryszard.piramidowicz@pw.edu.pl (R.P.); 2CEZAMAT, Warsaw University of Technology, 02-822 Warsaw, Poland; 3VIGO System S.A., 05-850 Ożarów Mazowiecki, Poland; 4TMS Diagnostyka Sp. z o.o., 15-276 Białystok, Poland; krzysztof.matuk@tmsdiagnostyka.pl; 5SMART Photonics B.V., 5656 AE Eindhoven, The Netherlands; luc.augustin@smartphotonics.nl

**Keywords:** interrogator, FBG, photonic integrated circuit, sensor network

## Abstract

In this work, we discuss the idea and practical implementation of an integrated photonic circuit-based interrogator of fiber Bragg grating (FBG) sensors dedicated to monitoring the condition of the patients exposed to Magnetic Resonance Imaging (MRI) diagnosis. The presented solution is based on an Arrayed Waveguide Grating (AWG) demultiplexer fabricated in generic indium phosphide technology. We demonstrate the consecutive steps of development of the device from design to demonstrator version of the system with confirmed functionality of monitoring the respiratory rate of the patient. The results, compared to those obtained using commercially available bulk interrogator, confirmed both the general concept and proper operation of the device.

## 1. Introduction

In contemporary medicine, there is an obvious, clearly visible trend in the continuous improvement of the quality of medical care by introducing various advanced technology-based diagnostic, monitoring, and life-support systems, which may improve the quality and efficiency of diagnosis, therapy, and patient care. To enumerate a few of these–a sophisticated and expensive medical equipment deployed in intensive care units for vital signs monitoring (like respiratory monitoring, pulse oximeters, cardiac monitors, hemodynamic monitoring) should be mentioned, together with advanced (and even more costly) diagnostic tools like ultrasound devices, computer tomography (CT), magnetic resonance imaging (MRI), positron emission tomography (PET), etc. The omnipresence of advanced technologies is evident. However, in typical everyday patient care, these are not commonly deployed, due to the lack of comfort, high cost, dimensions, limited availability of appropriate devices or sometimes distrust of modern technologies. Typically, numerous nursing staff takes care of the patients, periodically monitoring their parameters like temperature, blood pressure, blood oxygen saturation level, heart rate (HR), respiratory rate (RR), etc. Such monitoring is necessary, in the case of in-patients where there is also a number of medical examinations during which continuous monitoring of patient’s vital signs is highly desired, and specifically when the general condition of the patient is unstable or unknown (which applies to e.g., unconscious patients). An example might be an MRI scanning procedure, a non-invasive technology that enables obtaining three-dimensional anatomical images of exceptional resolution by the use of detecting the changes of protons energy in the presence of strong magnetic field and radiofrequency waves pulsed through the patient.

During such an examination, patients may experience discomfort due to a claustrophobic environment of the MRI measuring chamber, which typically has a form of a relatively tight tube, with the bed for the patient in the center. The dimensions and shape of the chamber might result in anxiety, claustrophobia, and panic attacks, specifically when the scan requires several or a few tens of minutes to be taken properly. This can potentially lead even to deterioration of the patient’s health. It should also be noted that any patient’s movements (e.g., due to an uncomfortable environment) inside the capsule might worsen the image quality or result in the necessity of a repeated examination. Therefore, monitoring the patient’s condition is highly desirable–the symptoms of increasing stress can be detected based on increased heart rate and/or respiratory rate, thus giving feedback to the medical personnel, who can react accordingly. The monitoring is even more important in the case of unconscious patients, who are not able to provide the staff with information on their actual condition.

Due to the presence of a strong electromagnetic field inside the MRI chamber, inherent to this imaging technique, it is impossible to use standard electronic sensors for monitoring the patient’s condition. Typically, MRI-certified electronic monitors are used, carefully shielded from the electromagnetic fields. An interesting alternative, however, may be offered by optical sensors, and specifically fiber-optic sensing elements like fiber Bragg gratings (FBGs) [1,2]. The investigations shown in [3,4,5,6,7,8,9,10] have proved the possibility of detecting heart rate (HR) and respiratory rate (RR) using optical signals reflected from FBGs, exposed to indirect contact with a human body. It has been confirmed that both HR and RR signals can be extracted from changes of the Bragg wavelength, which are the result of changes of the pressure imposed on FBGs by a human body, and which can be recorded using an appropriate interrogating system. We have successfully demonstrated such a system at Photonics Europe conference in Strasbourg, 2018 [11] equipped with a commercially available, bulky interrogating device. In this work, we present the progress in developing the interrogating part of the system, the core of which is realized as an application specific photonic integrated circuit (ASPIC). As integration technologies in photonics have significantly matured within the last years, nowadays access to silicon, silicon nitride, and indium phosphide platforms is offered on a commercial basis. The latter seems to be highly attractive as it provides monolithic integration of active and passive components, thus, enabling the design and fabrication of complex miniaturized, photonic systems, comprising coherent and broadband light sources, optical amplifiers, light modulators, and detectors. Furthermore, the integration of many functions in a single device significantly minimizes the fabrication and exploitation cost and increases the reliability of the final product [12,13,14].

The potential of the InP-based integration technology has been used to develop an integrated spectrometer device, comprising a wavelength demultiplexer, an array of photodiodes, and an optical amplifier, the combination of which enables effective interrogation of a fiber-optic sensor system based on Bragg gratings. In this work, the ASPIC design and the obtained characterization results are presented. System-level test results are presented and discussed with respect to the interrogation of a single FBG, as well as to detect the respiratory rate of a person lying on a bed with a mounted sensor plate.

## 2. Interrogator Design and Implementation

### 2.1. General Concept of the System

The monitoring system consists of two major parts–a network of FBG sensors integrated with MRI bed overlay and an interrogator located outside the measurement chamber connected to the sensors using a standard single-mode fiber. The system is depicted in Figure 1. The FBG sensors and optical fibers by their nature are insensitive to the magnetic field, so they can be installed inside an MRI chamber, while the entire interrogator system and computer for data analysis can be located outside, in the MRI operator room. Designed in such a way, the system is invisible to a patient (the measurements do not require direct contact with the patient’s body), and thus, does not reduce his/her psychical comfort. The sensors are also invisible to MRI detectors, and thus, do not influence the recorded image quality, which is even more important.

The heart of the system is the interrogating device, which is responsible for the proper detection of signals reflected from FBG sensors. Considering the intended use of the device, it should be highly reliable, preferably compact and cost-optimized. All the above-defined properties are characteristic of photonic integrated circuits (PICs). Therefore the investigations described in the following part were focused on this specific approach.

### 2.2. Integrated Interrogator Design

The commercially available devices usually exploit a read-out scheme basing on a broadband optical signal, e.g., from a superluminescent diode (SLED) source, launched to the sensors network, reflected, and analyzed with an optical spectrometer. In the alternative approach, a tunable monochromatic light is used, while the reflected signal is monitored with a single broadband photodetector. This enables the reconstruction of the FBG spectrum by analyzing the recorded signal in the time domain. In this work, the first approach was implemented. A general concept of the readout system for an FBG sensor network based on a broadband light source and an integrated wavelength filter–a demultiplexer with dedicated photodiodes is shown schematically in Figure 2.

To miniaturize the system shown in Figure 2, a single photonic integrated circuit (PIC) was designed equipped with an arrayed waveguide grating (AWG) demultiplexer with outputs connected to multiple PIN photodiodes. The topology of a standard AWG (originally dedicated to telecom applications) was used and two integrated interrogators differing in the spectral resolution were placed on a single chip designed for generic indium phosphide (InP) technological platform offered by SMART Photonics [15].

Both interrogators were designed as 36-channel devices, differing only in channel spacing, which was 50 GHz (which corresponds to 0.4 nm) and 75 GHz (0.6 nm), respectively. The free spectral range (FSR) of AWG was chosen as the channel spacing multiplied by the number of the AWG channels. Therefore, a 50 GHz channel spaced AWG was designed with an FSR of 14.4 nm and a 75 GHz one with an FSR of 21.6 nm. The chosen channel spacings of AWG were a trade-off between the readout device resolution and reasonably small chip size. The layout of the chip and its photograph are shown in Figure 3.

Both designed interrogators share the same general layout scheme. Each of them is equipped with two optical inputs, using angled waveguides. Out of two inputs the one is directly coupled to the AWG and the other one is connected via a build in semiconductor optical amplifier (SOA) to increase the power level of the input signal. The output waveguides of the AWG are connected to an array of PIN photodiodes. The layout is compatible with a CORDON generic packaging scheme, having the electrical interface at the bottom, the top and the right side of the chip, and optical inputs/outputs at the left side (see Figure 3).

## 3. Interrogator Validation

### 3.1. Characterization Results

Fabricated circuits were carefully characterized and tested with respect to the design parameters. The experimental setup consisted of a tunable laser source covering the spectral range 1545–1570 nm, a 3-D translation stage enabling precise positioning of a tapered optical fiber against the input waveguide, a chip temperature control unit, and a source measure unit (SMU) for driving the SOA, biasing the photodiodes and measuring their photocurrents. The tunable laser and PIC were connected using polarization-maintaining (PM) optical fibers.

The recorded transmission spectrum of 50 GHz AWG is presented on the left side of Figure 4, while a spectrum of AWG’s two adjacent channels is shown on the right side of Figure 4.

The measured values of FSR and channel spacing (average 14.4 nm and 0.4 nm respectively) are in good agreement with the design values. The device exhibits proper operation as a WDM demultiplexer having crosstalk better than 17 dB. For AWG’s central wavelength (λc = 1561.6 nm) the optical power should reach its maximal value. The side channels were expected to have lower peak values according to Gaussian distribution and this tendency is clearly seen in Figure 4. This type of transmission spectrum is characteristic for typical AWGs [16].

The use of a built-in SOA allows increasing the level of the photodiode current signal by 10 dB while maintaining other parameters of the AWG transmission spectrum unchanged. Figure 5 presents a transmission spectrum obtained for 16 channels of the 50 GHz AWG, when an optical signal is coupled to the input equipped with an SOA, driven with 70 mA.

Due to the smaller channel spacing of the 50 GHz AWG interrogator, and hence the better resolution of the reconstruction of the FBG reflection spectrum, this particular AWG was selected for the subsequent work. As the signal level in the interrogator without the use of SOA was satisfactory for this particular application, the SOA has not been used in the following experiments.

### 3.2. Interrogation of FBG Sensors

#### 3.2.1. Experimental System Configuration

To evaluate the operation of the interrogator in conditions equivalent to operational, a measurement system shown in Figure 6 was set up.

The SLED (Dense Light DL-BP1-CS5169A) equipped with a built-in optical circulator was used as the broadband light source. The single mode fiber (SMF)-based (type SMF-28) FBG single sensor/network of sensors was connected to one output of the light source, while the other port was connected via polarization control elements to the integrated interrogator (denoted as “device under test”-DUT) and to a reference device (REF)-a commercially available FBG interrogator (Ibsen I-MON 256 USB) or an optical spectrum analyzer (OSA, Yokogawa AQ6375, Yokogawa, Japan). The photodiodes of the integrated interrogator were connected to a source measurement unit (SMU, Keithley 2602B, Cleveland, OH, USA), which was used to provide reverse bias to the diodes and monitor photocurrent.

In our previous work [9], we have shown that the operation of the AWG demultiplexer is considerably polarization-sensitive, which may lead to incorrect wavelength readout when the input polarization state is unknown, which is a typical case of SMF-based fiber-optic systems. As there were no polarization-management on-chip components offered in available generic PIC technologies, the mitigating off-chip solution based on a fiber-based polarization beam splitter (PBS) and a 2 × 2 polarization-maintaining (PM) fiber coupler was introduced to eliminate the negative influence of the unknown polarization state of the input broadband optical signal.

#### 3.2.2. FBG Reflection Spectrum Reconstruction

In the first experiment, the possibility of properly reconstructing the shape of the FBG wavelength response was evaluated. An FBG with FWHM equal to 3.0 nm was used, which is significantly larger than the value of the channel spacing of the AWG (50 GHz/0.4 nm). The experiment was conducted by reading out the response at 14 consecutive AWG channels and comparing the result with the reflection spectrum collected with an optical spectrum analyzer. The obtained spectra are shown in Figure 7.

The measurement results confirm the possibility of the proper reconstruction of the FBG reflection spectrum with an AWG-based interrogator. The shape of the FBG response collected with the OSA corresponds well to the received signal of consecutive PIC outputs, however, a small shift (of the order of 0.5 nm) towards longer wavelengths is clearly seen. Such a shift, resulting from the technological imperfections of AWG is not critical as in principle the interrogator does not monitor the exact value of the wavelength, but its relative change. It is also noticeable that the signal-to-noise ratio (SNR) collected with OSA is about 15 dB larger than SNR in the case of the signal collected with an AWG. The smaller SNR measured with AWG is caused by the inter-channel crosstalk of AWG. As the AWG channel spacing (0.4 nm) is significantly larger than the wavelength scan step of the OSA (0.01 nm), each measurement point collected with AWG can be considered as averaging of 40 neighboring points collected with OSA. Consequently, the FBG response measured with AWG is smoother than the one collected with OSA.

#### 3.2.3. FBG Elongation Testing

The interaction of an AWG-based interrogator with a destined 0.7 nm FWHM FBG was evaluated in the experiment, where the 27 cm long section of SMF with an FBG inscribed was mounted in a translation stage allowing fiber tensioning and loosening with a precision better than 50 µm. Figure 8 shows the photocurrents of four consecutive photodiodes of the 50 GHz AWG -based interrogator during a performed measurement cycle of fiber tensioning and loosening. In relation to the diodes description, all diodes of the interrogator were named in alphabetic order compatible with electrical connectors defined by packaging standard, however, for this analysis, only those mentioned in Figure 8 will be used, due to the compatibility of spectral ranges.

At the beginning of the measurement cycle, the fiber was completely loose. It was subsequently tensioned step by step until the risk of fiber break occurred. The fiber was then relaxed until completely loosen. Then, the cycle was repeated. Finally, in order to determine the dark current of each photodiode, the optical signal was switched off completely, and the dark current was recorded.

With the elongation of the FBG, its wavelength response shifts towards longer wavelengths. Consequently, the optical signal coupled to the AWG-based interrogator is redirected from one interrogator receiving diode to a neighboring one, accordingly with longer wavelengths reflected from FBG. This phenomenon is depicted in Figure 9, matching a close-up on the experiment shown in Figure 8 with reflection spectra recorded with a reference interrogator.

During the cycle, the results of which are shown in Figure 9, the FBG was elongated by approximately 147 µm, thus, inducing a strain applied to the grating of approximately 545 µε (microstrains). Consequently, the wavelength response of the FBG was shifted by 0.60 nm (from 1532.90 nm to 1533.50 nm). At the starting point of the measurements, diode M receives the highest incoming signal. Then, as the grating elongates, the Bragg wavelength is shifted and the increasing signal is received by the diodes O, P, and Q (in that sequence), which is clearly seen in Figure 9a. It should be noted, that as the FWHM of FBG is comparable with the channel spacing of AWG, a considerable fraction of the energy is directed to neighboring channels and is detected by the corresponding diodes. For the reference the reflection spectra corresponding to the applied tensions (tension A, B and C) were recorded with the reference interrogator, connected to the optical fiber circuit as it is shown in Figure 6 (described as “REF”). The recorded reference spectra are shown in Figure 9b.

Having confirmed the general properties and proper response to mechanical stress applied to an FBG sensor, the more sophisticated version of the system has been developed, and first attempts to obtain a pattern of human breathing have been made. The developed system, designed and configured analogous to the setup shown in Figure 6, consists of a dedicated driver instead of SMU, an integrated interrogator packaged using CORDON generic packaging technology and a set of FBG sensors mounted on a small PMMA plate and installed in the bed overlay.

#### 3.2.4. Electronic Readout Circuit Design and Implementation

The dedicated electronic driver, shown in Figure 10 enables simultaneous processing and collection of all 36 signals from the photonic circuit electric ports. Its architecture is based on an STM32F303ZDT6 microcontroller, equipped with analog front-end circuitry and analog-to-digital converters (ADC). A current signal from each photodiode of the interrogator is amplified and converted to the voltage signal using a trans-impedance amplifier (TIA) and processed with a low-pass filter. The output signal is then subsequently digitized by an ADC. Finally, all the recorded signals are exported in real-time mode together with a timestamp to the PC via USB interface. The PCB layout enables direct assembly of a packaged PIC. Figure 10 presents a functional block diagram of the dedicated electronic driver with a packaged interrogator chip mounted on it. The approximate power consumption of the packaged device is 1.3 W.

#### 3.2.5. Human Respiratory Rate Detection Testing

To detect respiratory rate a fiber Bragg grating was mounted on a polymer board, which was placed between the patient’s back and the mattress, similar to [17]. This solution protects the sensor against unintentional displacement and damage.

Prior to starting the experiments with recording human breathing, the influence of the temperature of the human body has been analyzed. To distinguish the impact of the temperature on the detected signals an experiment with a heating pad placed on a sensing plate was performed, the results of which are depicted in Figure 11.

At the beginning of the measurement, the FBG plate was conditioned for 5 min at room temperature. After that, a heating pad filled with water of a temperature of 35.7 °C was placed on the sensor plate, which resulted in heating the sensors, and in turn, in a continuous shift of the Bragg wavelength towards longer wavelengths. After around 20 min the temperature of the FBG sensors (which went up during this time) and the heating pad (which went down) equalizes. Then, as the heating pad temperature consecutively decreases, the FBG temperature begins to decrease, so the Bragg wavelength starts to shift towards shorter wavelengths. At the 57th min of the measurement, the heating pad was removed from the sensor plate, which then started to cool down more quickly. Such changes of the temperature can be clearly visible in the time domain characteristic, presented in Figure 11b, measured using a reference FBG interrogator.

The analysis of the response of the photodiodes is more complex. Signals from diodes F and I, corresponding to the shortest wavelengths decrease, due to the shifting of the FBG response towards longer wavelengths. With FBG plate warming, the adjacent photodiodes (M, O, and P) signals are reaching their maxima successively. Diode Q, which is the next after P, detects some signal change, but due to its malfunction, is responding only partially. The next diodes (R, S, and T) also receive minor voltage changes consistent with temperature increase, but the FBG response is too narrow to have a considerable impact on their signal. After the FBG reaches the maximum temperature (around 20th min) and starts to cool down, the spectrum of the reflected signal shifts back to shorter wavelengths, which is visible in a weaker response of diodes P-T. Optical power is transferred back to the diode O and then M. When the heating pad is removed the FBG starts to cool down even faster, thus the diode O signal decreases rapidly while diode M reaches a maximum for a second time. When the FBG reaches back the room temperature the signals from diode I and then F increase, similar to the initial part of the measurement. It can be observed that the voltage curves recorded at the beginning and at the end of experiment differ slightly. The possible explanation is that the temperature conditions varyied during the experiment (which is also confirmed by reference interrogator). Nonetheless, it does not affect the ability to proper detect respiratory rate.

Finally, the experiments on detecting the breathing activity of a real person lying on the bed equipped with the FBG sensor plate were performed, the results of which are presented in Figure 12 and Figure 13. In Figure 12 the influence of two factors is clearly seen–the temperature of the human body discussed above and mechanical stress resulting from the movements of the chest while breathing. During the first 5 min, the FBG remained relaxed at room temperature on an empty mattress. The person then lay down on the bed, which introduced a temporary, rapid change of the interrogator diodes response due to the body movement. Slowly varying global shift of the response of the photodiode due to increase of the temperature stabilizes around 20th min of measurement when the FBG reaches the temperature of the body. The response of the system remained stable until the 35th min when the person got up. As expected, changes caused by chest movements are no longer observed. The relaxation of the FBG after the impact of the body disappeared results in a rapid change of the Bragg wavelength. As the temperature decreases, the slowly varying response of the diodes follows the same pattern as in the previous experiment. Similar to the previous measurement a reference interrogator was also used, the response of which shows exactly the same tendency of the Bragg wavelength changes. In the fifth min, the device registered a moment when a person laid on the mattress, followed by an increase of sensor temperature–a shift to longer wavelengths due to heating by the human body with clearly visible periodic breathing pattern. The last part of the trace corresponds to standing from the bed and stabilization of the sensor temperature at room temperature (RT) value.

It can be noticed that stabilization time is considerably long (ca. 20 min). However, it should be noted, that the reliable RR signal can be recorded irrespective of the temperature-induced wavelength change (which is slow-varying component comparing to the RR signal and can be easily removed numerically).

Figure 13 presents a five-minute-long zoom-in of the time traces presented in Figure 12, recorded after the temperature of the grating stabilizes. Throughout the measurement cycle, the person was breathing steadily at a natural pace, with a breathing period of c.a. 6 s. The chest movements induced stress on the FBG, which in turn, caused periodical variation of the Bragg wavelength, clearly visible both on the reference characteristic (the average period equal to 6.1 s) and on the signals from the photodiodes. Diodes F, I, and M are responding in this same pattern. The signals of the diodes F and I are very weak, as the FBG spectrum lies outside their wavelength channels. Diodes O, P, R, S, and T work in antiphase to previously described diodes. The strongest response is observed for I, M, and O diodes, as they cover the peak of the signal reflected from the FBG. Diode Q is not responding, due to its malfunction, mentioned above. It is seen that the breathing period detected by the PIC and the reference interrogator are very well matched.

The last measurement was made with the person lying on the mattress and breathing at a different rate for 3 min. Initially, the FBG plate was at room temperature and conditioned on an empty mattress. The volunteer then laid down and was breathing at a normal rate (1 breath for 5–6 s) for 90 s, then held his breath for 15 s (apnea), then began to breathe rapidly (1 breath for 1–2 s) for the next 20 s. Then he calmed his breathing and returned for the next 60 s to the pace with which he had started, and then he left the bed. The time traces recorded during this measurement are shown in Figure 14.

The corresponding patterns may be clearly distinguished with respect to the characteristics recorded using both devices. In addition to the breathing pattern, the graph shows the global effect of the FBG heating from the human body, manifested by an increase of the Bragg wavelength. This effect is noticeable even in a relatively short section of the apnea. The disturbance recorded by both devices in ~62 s of the measurement was the result of the patient’s movement. Due to a slight change in body position, the tension on the FBG changed, which is also visible in the graph. It should be noted that any physical movements do not influence the breathing pattern recorded. Moreover, since the MRI examination should be carried out with the patient body immobilized, the possibility of detecting the patient’s movements may be considered as an added value of the system, which could enable informing the technician to repeat the procedure.

## 4. Conclusions

In this paper, we have presented the compact optoelectronic system for monitoring fundamental vital signs of the patient under MRI scan, based on FBG sensors and photonic integrated circuit as an interrogating device, with so-far demonstrated functionality of the proper detection of the respiratory rate, while the work on heart rate detection is still in progress. The heart of the system is 36 channel AWG-based spectrometer, designed and manufactured on the indium phosphide generic platform provided by SMART Photonics company and driven by a dedicated, in-house developed electronic circuitry. The experiments confirmed the applicability of the developed system to targeted application–the PIC-based interrogator reacts properly to the changes of pressure caused by the breathing person’s chest to the FBG sensors and the temperature changes resulting from the human body warmth. Signal interrogation is provided with time resolution sufficient to track the speed of breathing and precisely distinguish between individual breaths. The developed device offers all the advantages inherent to photonic integrated circuits-high reliability, due to the elimination of volumetric elements, small size, low power consumption, and low cost, assuming a sufficiently large scale of manufacturing.

Although the system has been originally designed for monitoring the condition of the patients exposed to MRI diagnosis its features might be effectively used in other areas of monitoring the condition of the patients, for example in intensive care units, nursing homes, or private houses (e.g., for monitoring the quality of sleep, preventing the incidents of sleep apnea). Complementing the functionality with the heart rate (HR) monitoring will further increase the usefulness of the device and open the gate to market deployment. The implementation of this functionality would require the sampling frequency to be increased (the presently used electronic circuitry was too slow) and for the the detection algorithms to be upgraded. As it is a subject of current development, it will be reported in future papers.

## Figures and Tables

**Figure 1 sensors-21-04238-f001:**
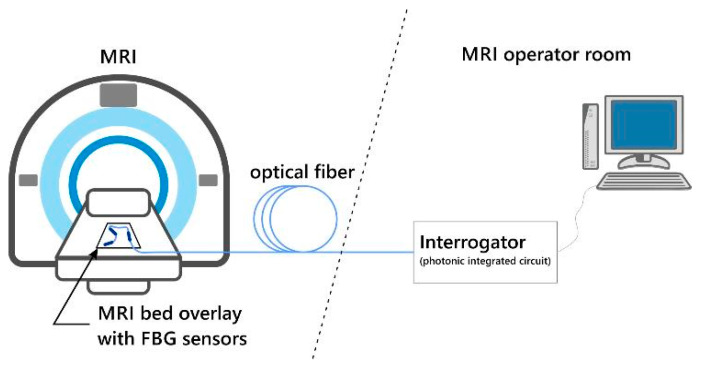
The general idea of the system for monitoring the vital functions of a patient undergoing an MRI procedure.

**Figure 2 sensors-21-04238-f002:**
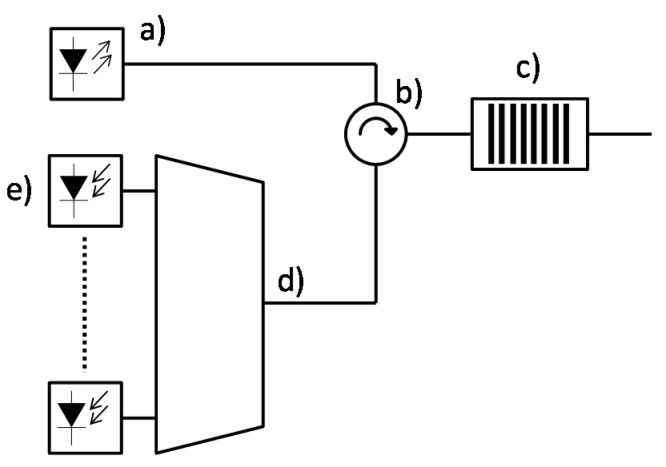
Readout system for FBG sensor network: (**a**) Broadband light source, (**b**) circulator (**c**) optical fiber with FBG sensors, (**d**) spectrometer, (**e**) array of photodiodes.

**Figure 3 sensors-21-04238-f003:**
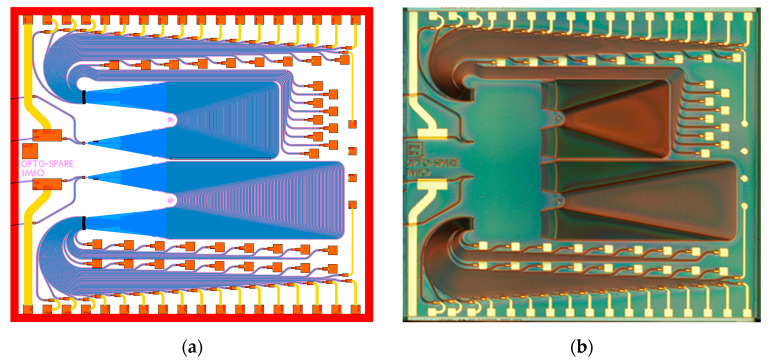
(**a**) Layout of PIC containing two integrated interrogators based on 36-channel AWGs having 75 GHz (top) and 50 GHz (bottom) channel spacing; (**b**) optical micrograph of a fabricated optical interrogator circuit.

**Figure 4 sensors-21-04238-f004:**
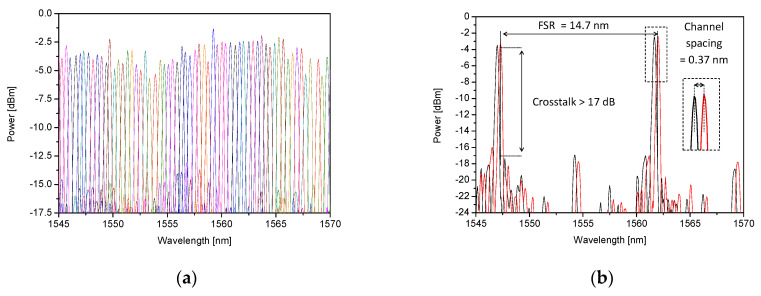
Transmission spectrum of (**a**) 36-channel; (**b**) two adjacent channels of 50 GHz channel-spaced AWG based interrogator.

**Figure 5 sensors-21-04238-f005:**
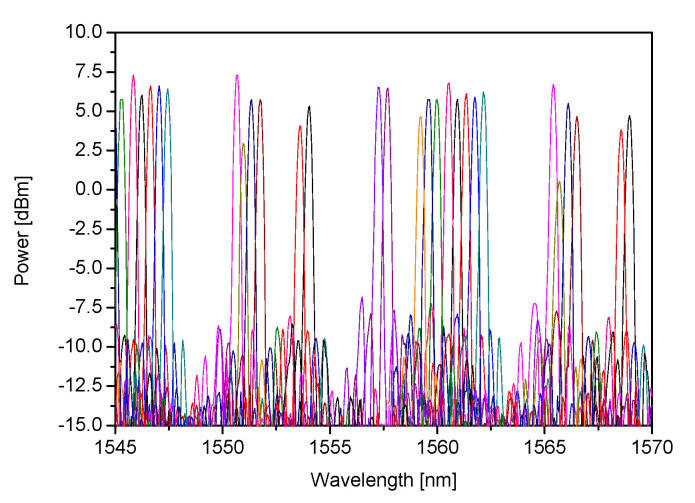
Transmission spectrum of 16 channels of 50 GHz channel-spaced AWG based interrogator with an active SOA.

**Figure 6 sensors-21-04238-f006:**
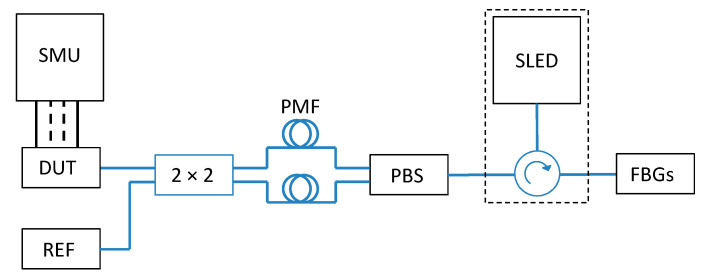
Schematic representation of the measurement system incorporating PIC-based interrogator.

**Figure 7 sensors-21-04238-f007:**
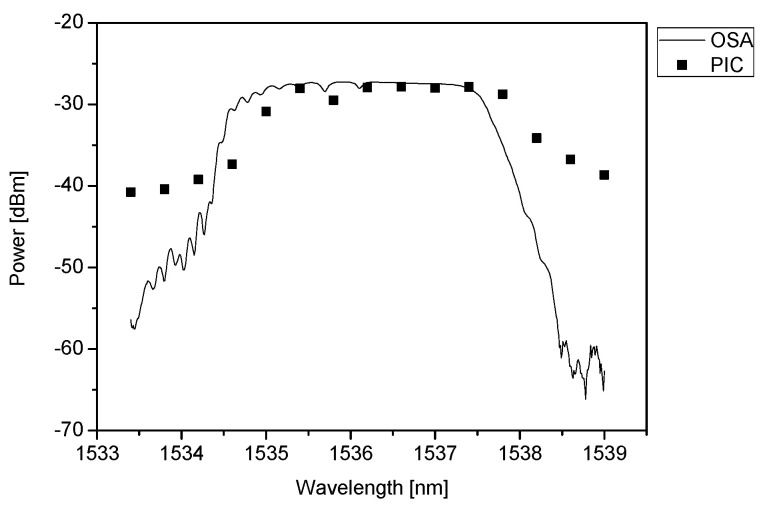
Comparison of the shape of FBG reflection spectrum collected with optical spectrum analyzer with the response of 14 consecutive photodiodes of the 50 GHz channel-spaced AWG-based interrogator.

**Figure 8 sensors-21-04238-f008:**
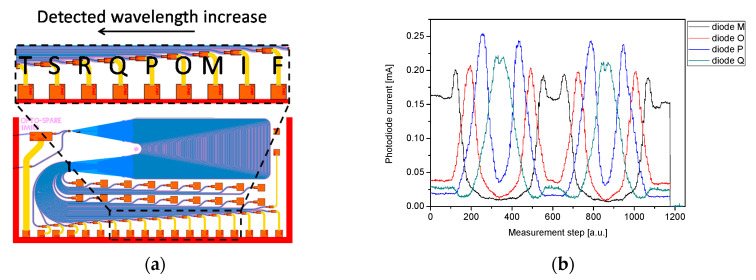
(**a**) Photodiodes names description on layout; (**b**) photodiode current measured during the experimental two cycles of fiber tensioning and loosening.

**Figure 9 sensors-21-04238-f009:**
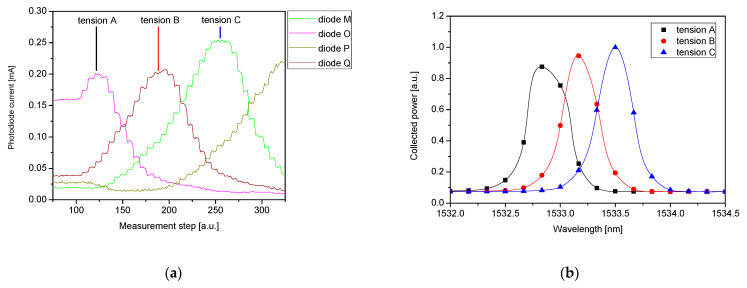
(**a**) Close up on the experiment shown in Figure 8, (**b**) corresponding reflection spectra recorded with the reference interrogator at given tensions applied.

**Figure 10 sensors-21-04238-f010:**
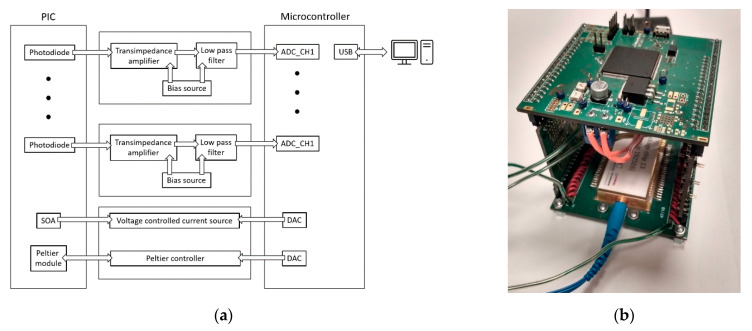
(**a**) Functional block diagram of the dedicated electronic driver; (**b**) assembly of the packaged interrogator chip.

**Figure 11 sensors-21-04238-f011:**
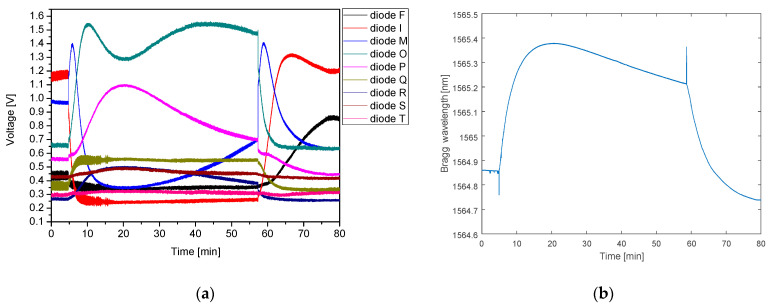
(**a**) Influence of temperature on FBG and PIC interrogator response; (**b**) response of the reference Ibsen I-MON 256.

**Figure 12 sensors-21-04238-f012:**
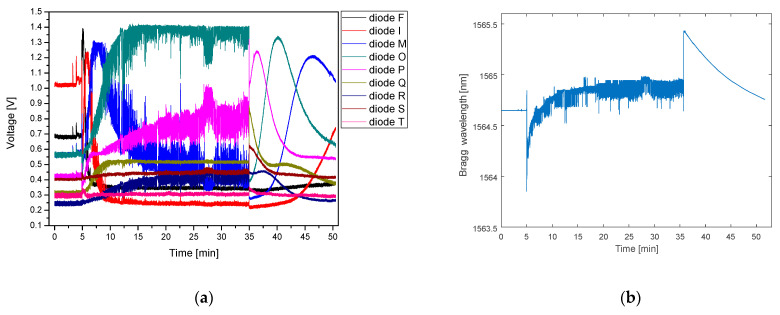
Regular breathing and effect of FBG heating up captured by (**a**) PIC interrogator; (**b**) Ibsen I-MON 256 interrogator.

**Figure 13 sensors-21-04238-f013:**
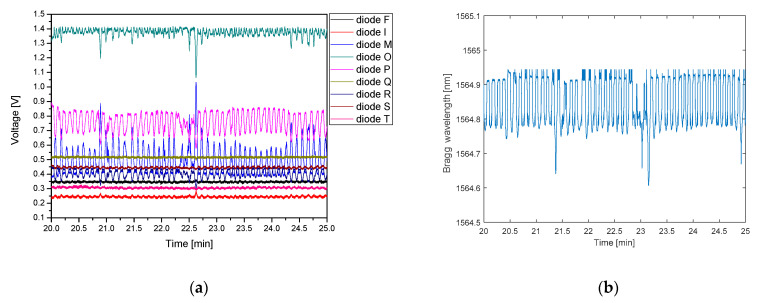
Previous measurement close-up with breath results captured by (**a**) PIC interrogator; (**b**) Ibsen I-MON 256 interrogator.

**Figure 14 sensors-21-04238-f014:**
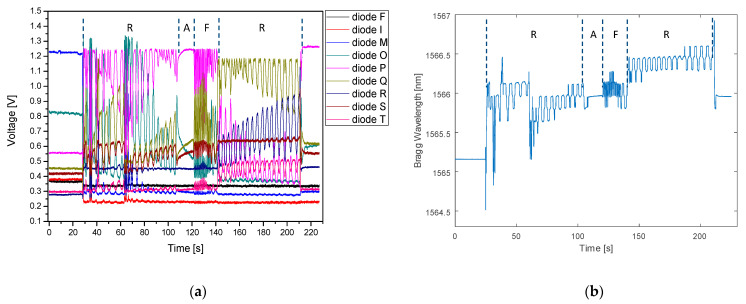
Result of measurement with person lying on the mattress and changing breathing pace: regular breathing (R), apnea (A), fast breathing (F) and regular breathing (R), captured by (**a**) PIC interrogator; (**b**) Ibsen I-MON 256 interrogator. Effect of FBG heating up is visible also.

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
