# Peer review of "Photonic Integrated Interrogator for Monitoring the Patient Condition during MRI Diagnosis"

_sensors, 2021, doi:10.3390/s21124238_

Round 1

Reviewer 1 Report

Here are my comments on the manuscript.

In general it is a nice work based on an idea to have a sensor that senses many things in the same time and place in small size. The authors have explained only the things that were the most important for them. BUt the paper is written for the readers who are intereseted and hopefully want to repeat such a research based on it. For that some details are missing and, I think the paper would be better with them.  Now come my momments in details.

line 186
Could be useful to mention what the abbrevation SMF is stands for (my guess: Single Mode Fiber) And the same is true for all the abbrevations in the text except for very the common ones like PMMA or PC

Figure 7
The figure caption should be together with the figure (at the same page under the figure)

line 201
Here starts the description of the experiments. There are no mention of the type or made of devices that were used.
For example there is no information abot the SLED, FBG, SOA. 
This way it is not really useful for anyone who want do some research based on this work. Please give these inportant details too.
If it is just a theoretical or conceptual work just write something like that:
"The SLED was ...W  ... type of... company but any other one with similar parameters can be used"

line 202
 At the first place where it is mentioned it should be given what OSA stands for (my guess: Optical Signal Analyzer) From that it is clear to anyone what does it means.

line 247
At the beginning we can see that the strain was 545 με. To me this unit is not really familiar. 
Generally epsilon stands for the quotient of absolute elongation and original length. I am afraid that other readers are also not familiar with this με unit

line 313
If I understand correctly at least 20 minutes needed to begin to measure the body temperature. I guess it is slow a bit.

A much better heat conducting (or thinner) protecting material should be used to reduce that "boot time". It would be better to mention what would you do to speed it up in the next version or future. It takes almost the same time as the MRI examination. To be inside for the double time that does not increase the comfort of the patient.

line 360
Such am unintended motion sensing could be advantegous since any movement of the body in the MRI device can decrease the sharpness of the image resulted. So the operator can be warned at too big movement and maybe the examination should be repeated.

Reviewer 2 Report

The authors present a photonic device able to track human breathing patterns with a suggested application for patients undergoing an MRI exam. The presented solution seems very interesting and clever. I have some suggestions to improve the paper:

  • Abstract: FBG acronym must be described.
  • 2.2. Integrated interrogator design: I would like to see more details about the device, as the  specifications or references of the fiber, the light source, the photodiodes, etc., so the paper could be reproducible by other labs.  
  • Figure 4: are (a) and (b) the same figure? The description on the text is confusing as well the caption of the figure. Also, the value for the FSR is 14.4 nm in the text and 14.7 in the figure.
  • 3.2. Interrogation of FBG sensors: the sections of the paper should be better organized. For example, there should be a break/division in this section when you switch to the human breathing test.
  • Figure 9 (b): should have different colors so they are not related with the same colors in (a). maybe the circle, square and triangle are already good markers to distinguish between curves? 
  • Figure 11, 12, 14 (a): is there a reason for the voltage curves to be different in the beginning and end of the experiment? Aren't the conditions the same? Also some colors are similar, is it possible to use for example different traces patterns, as '.-' or 'o-'?
  • Figure 14: it would be clear if the different breath patterns (apnea, fast breathing, etc.) are clear marked on the graph, perhaps with vertical lines? 
  • Conclusions: Is there a reason why the detection of the heart rate is not implemented? Maybe you could refer the difficulties of the implementation?

Thank you!
